# Prune at the Clients, Not the Server: Accelerated Sparse Training in Federated Learning

## Abstract

In the recent paradigm of Federated Learning (FL), multiple clients train a shared model while keeping their local data private. Resource constraints of clients and communication costs pose major problems for training large models in FL. On the one hand, addressing the resource limitations of the clients, sparse training has proven to be a powerful tool in the centralized setting. On the other hand, communication costs in FL can be addressed by local training, where each client takes multiple gradient steps on its local data. Recent work has shown that local training can provably achieve the optimal accelerated communication complexity (Mishchenko et al., 2022). Hence, one would like an *accelerated sparse training* algorithm. In this work we show that naive integration of sparse training and acceleration at the server fails, and how to fix it by letting the clients perform these tasks appropriately. We introduce Sparse-ProxSkip, our method developed for the nonconvex setting, inspired by RandProx (Condat & Richtárik, 2022), which provably combines sparse training and acceleration in the convex setting. We demonstrate the good performance of Sparse-ProxSkip in extensive experiments.

## 1 Introduction

Federated learning (FL) is a distributed machine learning approach that enables multiple edge devices to collaboratively train a shared model while keeping their data local (McMahan et al., 2017; Konečný et al., 2016; Bonawitz et al., 2017). This paradigm addresses significant privacy concerns by avoiding the need to transfer potentially sensitive data to a central server and thus can enable access to huge datasets. Instead, local models are trained on each client's device, and only the model updates are aggregated at the server to train a shared global model. However, one of the main challenges in FL is the limited computational and communication resources of edge devices (Caldas et al., 2018b).

Pruning is a well-known technique in the centralized setting for reducing the computational and memory costs of model training and inference (Han et al., 2015; Evci et al., 2020; Lee et al., 2024). There are two major directions: *dense-to-sparse* or *sparse-to-sparse* training (Liu & Wang, 2023). *Dense-to-sparse (DTS)* training starts with a dense network and proceeds by systematically removing redundant or less important parameters and reduces the model size without substantially sacrificing performance. *Sparse-to-sparse (STS)* training starts with a sparse network and usually proceeds by sparsifying and regrowing weights but keeping the sparsity constant. Both lead to computational savings at inference time as the final model is sparse (Srinivas et al., 2017). But sparse-to-sparse training also leads to substantially reduced training costs as the model is sparse throughout the whole process. Hence, a sparse-to-sparse algorithm for FL would address the resource limitation of edge devices for efficient training and inference.

However, a key issue during training in FL are communication costs, as for every step of the optimizer the clients have to share the model updates with the server or with each other. *Local training* has emerged as the key paradigm for efficient learning which allows the participating clients to take multiple update steps before communicating with each other. It first appeared in the popular algorithm FedAvg and showed great empirical success in applications (McMahan et al., 2017). In a recent breakthrough, Mishchenko et al. (2022) introduced ProxSkip, the first algorithm to be provably more communication-efficient than FedAvg by employing control variates and randomization.

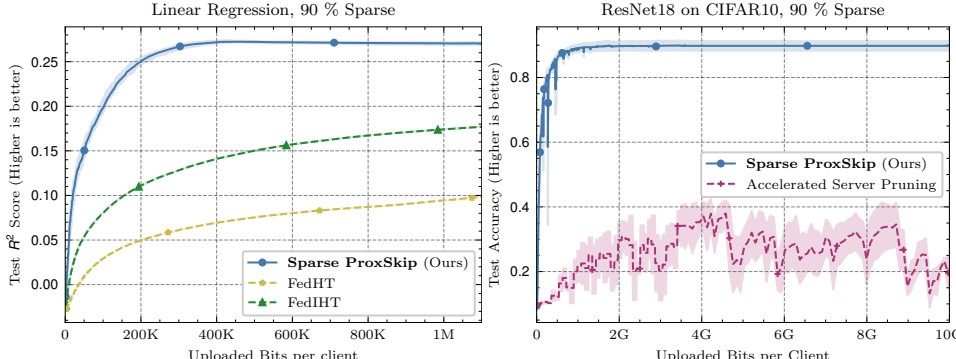

Figure 1: On the left, test score for regression on the Blog Feedback dataset (Buza, 2013). Our method performs best in both final score and communication efficiency. On the right, test accuracy for ResNet18 (He et al., 2016) on CIFAR-10 (Krizhevsky, 2009). Our method Sparse-ProxSkip prevents catastrophic failure occurring when combining acceleration and pruning at the server. The shaded area in both figures represents the standard error.

In a follow-up work, Condat & Richtárik (2022) were able to generalize the acceleration guarantees of ProxSkip to allow for multiple proxs in an algorithm called RandProx. In the convex setting with $l_1$ regularization, RandProx allows to obtain a sparse model while employing acceleration, although there is no guarantee on the sparsity level. However, in practice, $l_1$ regularization is usually outperformed by nonconvex techniques based on the $l_0$ seminorm.

**Challenge.** To achieve an efficient algorithm for FL, sparse-to-sparse training and the recent theoretical advances on acceleration need to be combined. Hence, we address the following research question:

*Is it possible to incorporate acceleration with nonconvex techniques usually found in sparse-to-sparse training algorithm?*

**Contributions.** A common approach in the FL literature is to apply pruning at the server (Stripelis et al., 2022; Lee et al., 2024). First, we show that this naive approach fails in the case of ProxSkip. Then, inspired bt RandProx, we derive a new algorithm, Sparse-ProxSkip, which addresses the problems by pruning at the clients instead of the server. We show that this is necessary through a combination of theory and experiments. Finally, we validate our algorithm in extensive experiments. Figure 1 shows how our proposed algorithm outperforms baselines for convex and deep learning experiments.

Hence, the paper starts with an overview of the theoretical background of RandProx and its application for pruning through $l_1$ regularization in section 3. Section 4.1 then shows the superiority of this algorithm for regression on the Blog Feedback dataset. Regression was chosen, as the theoretical guarantees hold only in convex scenarios like this and centralized STS regression is a well established field known as Subset Selection (Hastie et al., 2017). Section 4.2 establishes the superiority in logistic regression on FEMNIST (Caldas et al., 2018a). Finally, Section 4.3 deals with deep learning experiments.

## 2 RELATED WORK

Despite some existing studies on deriving sparse models in federated learning, the topic remains insufficiently understood. The most similar STS approach is given by Tong et al. (2020), who combine FedAvg and TopK to yield FedHT and FedIHT. Their approach does not integrate acceleration or control variates. Hence, this will be considered a baseline for our work. Furthermore, only FedIHT prunes the model before sending it to the server and thus uses the major communication efficiency of training a sparse model instead of a dense one (Yi et al., 2024). Subsequent works do not incorporate acceleration or address client drift either (Lin et al., 2022; Bibikar et al., 2022; Horvath et al., 2021; Isik et al., 2022; Tian et al., 2024; Huang et al., 2022), or they are not fully STS (Jiang et al., 2022; Qiu et al., 2022; Munir et al., 2021; Li et al., 2021).

In the DTS regime, the most simple approach is given by FedSparsify, which applies Gradual Magnitude Pruning in FedAvg at the server (Stripelis et al., 2022). The main difference between FedHT and FedSparsify is that the latter starts with a dense model and ramps up the sparsity by a cubic schedule during the training as is usual in centralized pruning. Another recent DTS work takes the approach of applying further centralized training approaches at the server (Lee et al., 2024). Here, one gathers up the local updates (usually with a fixed learning rate) and treats them as the gradient at the server. Then one can apply both centralized optimizers and centralized pruning techniques. In particular, Lee et al. (2024) apply the DTS techniques of random pruning, saliency pruning (Molchanov et al., 2016), GMP (Zhu & Gupta, 2017) and Straight Through Estimation (Bengio et al., 2013) and for STS they apply static sparse training, dynamic sparse training (Mocanu et al., 2018) and RigL (Evci et al., 2020). We will show that acceleration and pruning at the server fail and need to be applied at the clients instead. Hence, our work enables integrating all of the aforementioned centralized pruning techniques with ProxSkip or Scaffold (Karimireddy et al., 2020).

## 3    PROPOSED METHOD

Our algorithm is based on the recent progress in understanding local training made in Mishchenko et al. (2022). Their algorithm ProxSkip can optimize functions of the form

$$\min_{w \in \mathbb{R}^d} f(w) + \psi(w), \tag{1}$$

where $f$ is $L$-smooth and $\mu$-strongly convex and $\psi$ is proper, closed and convex (Bauschke & Combettes, 2017). It corresponds to Algorithm 1 with the pruning options disabled. Under these assumptions, the optimum $w^*$ exists and is unique. Hence, one can look at convergence against this optimum $w^*$. Let $w^0$ be the initial model estimate and $w^t$ be the iterate of their algorithm after $t$ steps. They proved that to be $\epsilon$ close to the optimum, i.e. $\|w^t - w^*\| \leq \epsilon \|w^0 - w^*\|$, one needs to evaluate the proximity operator (prox) of $\psi$ only $\sqrt{\frac{L}{\mu}} \log \frac{1}{\epsilon}$ times, while the best known bounds for Gradient Descent (and thus especially FedAvg) is $\frac{L}{\mu} \log \frac{1}{\epsilon}$. One main application of ProxSkip to FL is

$$\min_{w \in \mathbb{R}^d} \left\{ f(w) := \frac{1}{N} \sum_{i=1}^{N} f_i(w) \right\},$$

where $f_i : \mathbb{R}^d \to \mathbb{R}$ is the loss function of each client and $N$ is the total number of clients. This approach is closely related to empirical-risk minimization (Shalev-Shwartz & Ben-David, 2014), the dominant approach in supervised machine learning. In practice, $f_i$ is the individual loss function of Client $i$, based on their private and local data. This problem is a particular case of (1), using a consensus formulation (Parikh & Boyd, 2014). That is, the model $w \in \mathbb{R}^d$ is duplicated into $N$ independent copies $w_1, w_2, \ldots, w_N$ and the objective is changed to

$$\min_{w_1, \ldots, w_N \in \mathbb{R}^d} \frac{1}{N} \sum_{i=1}^{n} f_i(w_i) + \psi(w_1, \ldots, w_N),$$

where $\psi : (w_1, \ldots, w_N) \mapsto \{0 \text{ if } w_1 = \cdots = w_N, +\infty \text{ otherwise}\}$. The proper closed convex function $\psi$ encodes the consensus constraint and the theory of ProxSkip applies. The prox of $\psi$ $\text{prox}_{\gamma \psi}(w_1, \ldots, w_N) = (\bar{w}, \ldots, \bar{w}) \in \mathbb{R}^{Nd}$, where $\bar{w}$ is the average of the $w_i$. Thus, evaluating the prox boils down to communicating all local models $w_1, w_2, \ldots, w_N$ to a central server and averaging them. Hence, one prox evaluation corresponds exactly to one communication round, the main bottleneck in in FL (McMahan et al., 2017). Thus, reducing the number of prox evaluations is crucial to accelerate FL, which is why ProxSkip is such an important achievement for FL.

### 3.1    BASELINE METHODS

Additionally to FedHT and FedIHT discussed in the Section 2, we consider the following simple baselines of how to address the research question of incorporating pruning, acceleration and tackling client drift. A simple approach is to employ an accelerated algorithm like ProxSkip to obtain the dense solution $w^*$ and then take $\text{Top}K(w^*)$ of it for the desired sparsity, where the $\text{Top}K$ operator keeps the $K$ largest elements of a vector unchanged and sets the other ones to zero. This approach

does not address resource constraints of the clients or take advantage of training a sparse model to reduce communication cost. We will call this approach Final-TopK. The experiments will show that Sparse-ProxSkip addresses client resources and outperforms this method, showing that it provides a valuable contribution.

Another approach would be to consider pruning at the server, i.e. applying $\text{Top}K$ after averaging the model and before sending it back to the clients. Applying optimization techniques at the server is a common approach in FL (Lee et al., 2024; Lin et al., 2022; Stripelis et al., 2022). When applied to ProxSkip, we refer to this variant as Accelerated-Server-Pruning and it can be found in Algorithm 1. A major drawback is that this method does not benefit from compression for saving on uplink communication costs. As pruning is done before downlink communication, the models uploaded to the server are dense, incurring full communication cost. Furthermore, we show in the experiments that Accelerated-Server-Pruning violates a key invariant of control variates, so that it is essentially inappropriate for FL.

## 3.2 ACCELERATED PRUNING METHOD FOR FL WITH $l_1$ REGULARIZATION

Recently, Condat & Richtárik (2022) extended the framework of ProxSkip to allow for several proxs while keeping acceleration. In FL this means their algorithm RandProx can optimize problems of the form

$$\min_{w_1,\ldots,w_N \in \mathbb{R}^d} \frac{1}{N} \sum_{i=1}^{N} f_i\left(w_i\right) + \psi\left(w_1,\ldots,w_N\right) + h(w_1,\ldots,w_N),$$

for $h$ proper, closed and convex. One interesting case is to set $h(w) = \|w\|_1$, which comes down to *federated lasso* (Barik & Honorio, 2023). This model is known practically and theoretically to perform some sort of pruning, since it reduces the number of nonzero parameters (Barik & Honorio, 2023). Furthermore, the $l_1$ norm is convex, so that for convex loss functions $f_i$ the accelerated convergence guarantees of RandProx hold. We refer to this sparse training method as RandProx-$l_1$.

## 3.3 NONCONVEX MODIFICATIONS: CARDINALITY CONSTRAINTS

In practice however, it is well known that magnitude-based pruning methods have proven to outperform $l_1$ regularization, because of the bias it introduces. Cardinality constraints do not have this drawback and the algorithm can obtain the optimal solution on the subspace of the nonzero variables. Cardinality constraints can be represented in RandProx. One can set

$$h\left(w\right) := \begin{cases} 0, & \text{if } \|w\|_0 \leq K \\ +\infty, & \text{otherwise,} \end{cases}$$

where $\|w\|_0$ counts the number of nonzero components of $w$. RandProx makes calls to the prox of $h$, which is the hard-thresholding operator $\text{Top}K$ (Blumensath & Davies, 2009). The major caveat here is that this function $h$ is nonconvex, so that the proven acceleration guarantees of Condat & Richtárik (2022) do not hold. Empirically though, algorithms designed for the convex case have been proven powerful in the nonconvex case as well. So, we use the theoretical guarantees in the convex case as a strong guidance toward a powerful practical algorithm for the nonconvex case. The resulting algorithm is Sparse-ProxSkip-Local and it can be found in Algorithm 1.

## 3.4 FURTHER MODIFICATIONS AND PROPOSED ALGORITHM

A further improvement comes from the insight that one does not need to sparsify every step to gain a pruned model. The major beneficial sparsification happens before communication, as this reduces the communication cost similar to compression in FL algorithms (Condat et al., 2023; Yi et al., 2024). Hence, one could take the perspective of the models being pruned locally only due to the limited resources at the clients. To investigate the potential gains we consider a variant of the algorithm which only prunes before communication. The resulting algorithm is Sparse-ProxSkip found in Algorithm 1.

Here one has to take a choice where to place the $\text{Top}K$ operator. For the control variates $h_i$ to work properly, we show theoretically and empirically that it is crucial that $\sum_i h_i = 0$ always holds. For

---

**Algorithm 1** Meta Sparse-ProxSkip

---

1: stepsize $\gamma > 0$, probability $p > 0$, initial iterate $w_{1,0} = \cdots = w_{N,0} \in \mathbb{R}^d$, initial control variates $h_{1,0}, \ldots, h_{n,0} \in \mathbb{R}^d$ on each client such that $\sum_{i=1}^N h_{i,0} = 0$, number of iterations $T \geq 1$
2: **server:** flip a coin, $\theta_t \in \{0, 1\}$, $T$ times, where $\text{Prob}(\theta_t = 1) = p$
3: send the sequence $\theta_0, \ldots, \theta_{T-1}$ to all workers
4: **for** $t = 0, 1, \ldots, T - 1$ **do**
5:    **in parallel on all workers** $i \in [N]$ **do**
6:       $\hat{w}_{i,t+1} = w_{i,t} - \gamma(\nabla f_i(w_{i,t}) - h_{i,t})$       $\diamond$ SGD step adjusted by control variate $h_{i,t}$
7:       Option Sparse-ProxSkip-Local: $\hat{w}_{i,t+1} = \text{Top}K(\hat{w}_{i,t+1})$
8:       **if** $\theta_t = 1$ **then**
9:          Option Sparse-ProxSkip: $\hat{w}_{i,t+1} = \text{Top}K(\hat{w}_{i,t+1})$
10:          $w_{i,t+1} = \frac{1}{N} \sum_{j=1}^N \hat{w}_{j,t+1}$       $\diamond$ Communication with the server
11:          Option Accelerated-Server-Pruning: $w_{i,t+1} = \text{Top}K(w_{i,t+1})$
12:          $h_{i,t+1} = h_{i,t} + \frac{p}{\gamma}(w_{i,t+1} - \hat{w}_{i,t+1})$       $\diamond$ Update the local control variate $h_{i,t}$
13:       **else**
14:          $w_{i,t+1} = \hat{w}_{i,t+1}$       $\diamond$ Skip communication!
15:          $h_{i,t+1} = h_{i,t}$
16:       **end if**
17:    **end local updates**
18: **end for**
19: $w_{i,T} = \text{Top}K(w_{i,T})$

---

ProxSkip, one can now decide whether to apply $\text{Top}K$ before or after saving $\hat{w}$ at the clients. The change in Algorithm 1 is subtle. One takes Line 7 of Algorithm 1 to be either

$$\hat{w}_{i,t+1} = \text{Top}K(\hat{w}_{i,t+1}) \quad \text{or} \quad w_{i,t+1} = \text{Top}K(\hat{w}_{i,t+1}).$$

One can verify that our proposed variant, Sparse-ProxSkip, keeps the guarantee of $\sum_i h_i = 0$. Going back to ProxSkip, if this condition does not hold, one can show that the algorithm diverges. To see this, let us look at the simple case of $p = 1$ and $w_{i,0} = w^*$ for all $i$, i.e. just taking one local step located at the optimum. If $\sum_i h_i \neq 0$ then one gets at the aggregation step on the server

$$\frac{1}{N} \sum_{i=1}^N w^* - \gamma(g_i(w^*) - h_i) = w^* + \frac{1}{N} \sum_{i=1}^N \gamma(g_i(w^*) - h_i) = w^* + \frac{1}{N} \sum_{i=1}^N h_i \neq w^*.$$

The equality holds because $\sum_{i=1}^N g_i(w^*) = 0$ by first-order optimality conditions. Hence, $w^*$ is not a fixed-point and the algorithm diverges instead. We confirmed this hypothesis empirically for regression and logistic regression and provide a detailed analysis for logistic regression in Section 4.2.1.

## 4 EXPERIMENTS

We start with convex experiments for the following reasons. First, the convex setting is well understood and the theoretical guarantees of ProxSkip and RandProx hold only in this case. From a theory point of view, $\text{Top}K$ is not nonexpansive and hence might lead to divergence. Hence, we start with the convex setting to clearly investigate the effects of the mechanisms. Second, convex models are still surprisingly widespread in industrial applications. Third, many successful methods for the nonconvex case were designed for the convex case and then adapted to the nonconvex case. And lastly, ProxSkip and related accelerated methods are even without pruning still underexplored in the deep learning case. Hence, adapting these methods for sparse deep learning is challenging, but we provide experiments and general insights for this setting as well. General experimental details can be found in Appendix A.

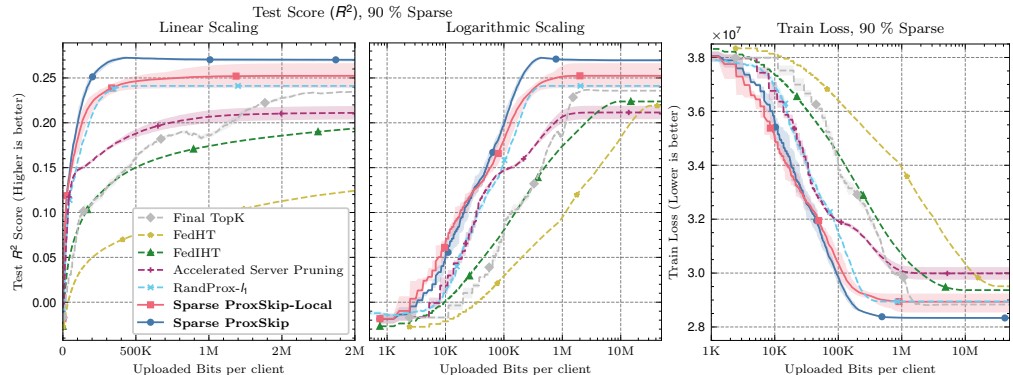

Figure 2: Test Score ($R^2$) on the left and train loss on the right for regression on the Blog Feedback dataset (Buza, 2013). Baseline methods are dashed while our methods are solid. One can observe that both RandProx-$l_1$ and our proposed methods converge to a better solution in a substantially more communication-efficient way. The shaded area in the figures represents the standard error. Error bars for all experiments are included but are sometimes not visible, due to deterministic initialization at $w_{i,0} = \mathbf{0}$.

Table 1: Multiple linear regression results. Sparse-ProxSkip shows an increase in $R^2$ due to addressing client drift. Table 5 (in the Appendix) additionally reports the final train loss. Results were obtained running a random search for $\gamma$ and $p$ for all algorithms.

|  | Sparsity | 80 % | 90 % | 95 % |
|---|---|---|---|---|
|  |  | Test $R^2$ | Test $R^2$ | Test $R^2$ |
| Existing | Final-TopK | 26.4 % | 23.8 % | 16.4 % |
|  | FedHT | 18.0 % | 21.9 % | 12.8 % |
|  | FedIHT | 16.5 % | 22.4 % | 12.3 % |
|  | Accelerated-Server-Pruning | 25.9 % | 20.4 % | 16.3 % |
|  | RandProx-$l_1$ | 26.3 % | 24.1 % | 18.8 % |
| Ours | Sparse-ProxSkip-Local | 27.0 % | 26.8 % | 23.9 % |
|  | Sparse-ProxSkip | 26.7 % | 27.1 % | 26.7 % |

## 4.1 MULTIPLE LINEAR REGRESSION ON BLOGFEEDBACK

**Setup.** The first experiments tackle multiple linear regression on the BlogFeedback dataset Buza (2013). We chose this dataset for providing a realistic example of a regression problem with a natural but challenging FL split. Previously, it has been used by Barik & Honorio (2023) to investigate federated lasso, which also addresses the challenge of feature selection in a federated regression problem. The total number of data points is $n = 47157$ split in a very heterogenous way across 554 clients. Furthermore, all results have been obtained by running a random search to tune the number of local steps $\frac{1}{p}$ and the learning rate $\gamma$. Error bars are obtained by running the same combinations 5 times for the same parameters with different random initialization if applicable. More details on the dataset and the experimental setup can be found in Appendix B.

**Experimental Results.** Our methods improves both in $R^2$ (quality of the solution) and in communication efficiency over the baselines. Training trajectories for a sparsity of 90% detailing the gains in communication cost and accuracy at the same time can be found in Figure 2. Table 1 reports the final $R^2$ (solution quality) for different target sparsity values. At 90% sparsity, we see that Sparse-ProxSkip improves by 3.2% over the best baseline Final-TopK and 6.6% over the best non client drift addressing variant. Furthermore, the advantage grows with increased sparsity at 95%. Table 2 reports the gains in communication efficiency. We can observe that Sparse-ProxSkip is roughly 10× more communication-efficient as the best baseline Final-TopK and roughly 20× more communication-efficient than the best non-accelerated baseline.

Table 2: Communication cost to reach a certain test $R^2$ score for multiple linear regression at $90\%$ sparsity. All speedup comparisons are with respect to Final-TopK as it is an accelerated method outperforming FedIHT and is the only baseline reaching a test score of $0.225$.

| | Test $R^2$ Threshold | 0.2 | | 0.225 | | 0.25 | |
|---|---|---|---|---|---|---|---|
| | Upload Communication Cost | Bits | Speedup | Bits | Speedup | Bits | Speedup |
| Existing | Final-TopK | 1.16 M | $1.00\times$ | 1.44 M | $1.00\times$ | ✗ | ✗ |
| | FedHT | 14.8 M | $0.08\times$ | ✗ | ✗ | ✗ | ✗ |
| | FedIHT | 2.49 M | $0.47\times$ | ✗ | ✗ | ✗ | ✗ |
| | Accelerated-Server-Pruning | 0.73 M | $1.59\times$ | ✗ | ✗ | ✗ | ✗ |
| | RandProx-$l_1$ | 0.18 M | $6.44\times$ | 0.25 M | $5.76\times$ | ✗ | ✗ |
| Ours | Sparse-ProxSkip-Local | 0.13 M | $8.90\times$ | 0.21 M | $6.86\times$ | 0.76 M | - |
| | Sparse-ProxSkip | 0.10 M | $11.6\times$ | 0.14 M | $10.3\times$ | 0.19 M | - |

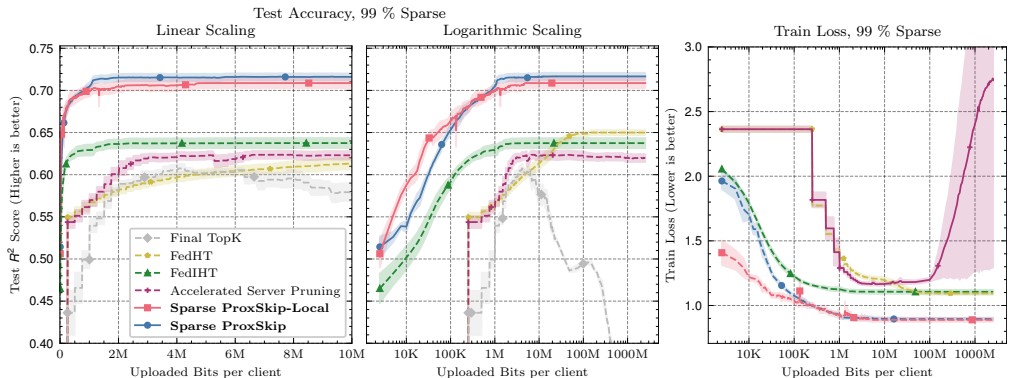

Figure 3: Results for logistic regression on FEMNIST at $99\%$ sparsity. Sparse-ProxSkip and Sparse-ProxSkip-Local outperform all baselines both in communication costs and final accuracy. The shaded area in the figures represents the standard error.

**RandProx-$l_1$ Beats Simple Baselines.** We see that RandProx-$l_1$, as described in Section 3.2, outperforms the simple baselines in terms of both communication efficiency and $R^2$.

Noticeably, this supports our hypothesis in that: 1) Acceleration (through RandProx-$l_1$) leads to a communication cost decrease of $\geq 6\times$ compared to FedIHT. 2) Addressing client drift (through RandProx-$l_1$) leads to an increase in final test score of up to $2.4\%$ compared to FedIHT. 3) RandProx-$l_1$ outperforms naive baselines like pruning at the server or pruning at the end, showing the need for a properly designed accelerated STS method.

**Failure of *Accelerated Server Pruning*.** From Figure 2 one can observe that Accelerated-Server-Pruning performs worst from all tested baselines. In particular, it performs worse than FedIHT which does neither address client drift nor is accelerated. As we discussed in Section 3.4 we hypothesized this because the property $\sum_i h_i \neq 0$ is violated in Accelerated-Server-Pruning. We confirmed this hypothesis empirically for logistic regression and provide a detailed analysis in Section 4.2.1.

**Sparse ProxSkip-Local beats RandProx-$l_1$.** We finally note that Sparse-ProxSkip outperforms RandProx-$l_1$ and the other baselines. We make the following observations: 1) RandProx-$l_1$ reaches the desired sparsity only gradually. The theory only guarantees convergence to a sparse solution, but there is no guarantee during the training. Hence, the communication costs it occurs are larger than when applying $\text{Top}K$ directly to sparsify locally. 2) One can notice an accuracy gain of Sparse-ProxSkip-Local compared to RandProx $l_1$. We attribute this to the bias induced by $l_1$ regularization.

Table 3: Communication costs to reach a certain test accuracy at 90% sparsity on FEMNIST. Note that although the final accuracy for Accelerated-Server-Pruning is below 80% as seen in Table 4, it peaks at 84 % early on. The same holds for Final-TopK and 85 %.

| | Test Accuracy Threshold | 80 % | | 82.5 % | | 85 % | |
|---|---|---|---|---|---|---|---|
| | Upload Communication Cost | Bits | Speedup | Bits | Speedup | Bits | Speedup |
| Existing | Final-TopK | 6.0 M | 0.1× | 16.6 M | 0.1× | 92.4 M | 0.1× |
| | FedHT | 4.0 M | 0.2× | 8.54 M | 0.2× | 45.7 M | 0.3× |
| | FedIHT | 0.8 M | 1.0× | 1.61 M | 1.0× | 13.0 M | 1.0× |
| | Accelerated-Server-Pruning | 2.0 M | 0.4× | 3.57 M | 0.5× | ✗ | ✗ |
| Ours | Sparse-ProxSkip-Local | 0.5 M | 1.8× | 0.55 M | 2.9× | 2.52 M | 5.2× |
| | Sparse-ProxSkip | 0.2 M | 4.0× | 0.35 M | 4.6× | 0.65 M | 19× |

Table 4: Test accuracy of logistic regression on FEMNIST for different sparsity levels. The best accuracy for each sparsity level is highlighted in bold.

| | Sparsity | 80 % | 90 % | 95 % | 98 % | 99 % |
|---|---|---|---|---|---|---|
| Existing | Final-TopK | 84.7 % | 79.9 % | 69.6 % | 40.1 % | 25.5 % |
| | FedHT | 86.6 % | 85.7 % | 84.7 % | 76.6 % | 66.4 % |
| | FedIHT | 86.8 % | 85.6 % | 82.7 % | 74.6 % | 65.4 % |
| | Accelerated-Server-Pruning | 77.9 % | 77.5 % | 76.8 % | 72.2 % | 64.7 % |
| Ours | Sparse-ProxSkip-Local | 86.7 % | 86.1 % | 84.7 % | 78.9 % | **72.9 %** |
| | Sparse-ProxSkip | **87.0 %** | **86.4 %** | **85.3 %** | **80.5 %** | 72.8 % |

## 4.2 Multiple Logistic Regression on FEMNIST

**Setup.** A more challenging but still convex setting is multiple logistic regression on the FEMNIST dataset (Caldas et al., 2018a). We take the naturally-occurring federated split but limit the number of clients to $N = 100$. A similar approach was taken by Jiang et al. (2022) for $N = 193$. The reasoning and further details can be found in Appendix C.

**Results.** The general results are shown in Figure 3. Results on communication efficiency are reported in Table 3. As only FedIHT enjoys communication speedup from compression, it is taken as the baseline so that the reported speedup is solely due to acceleration. We see that Sparse-ProxSkip-Local is 1.78–5.18× more communication-efficient and Accelerated-Server-Pruning is 4–20× more communication-efficient than FedIHT. If FedHT is taken as the baseline, which would be a usual approach for obtaining pruned models in FL (Lee et al., 2024), then Sparse-ProxSkip-Local is 9–18× and Accelerated-Server-Pruning is 20–70× more communication-efficient than FedHT.

Results on the final accuracy for different sparsity levels are reported in Table 4. One can observe that the advantage of our method is significant only with high sparsity levels. That is, at 80 % there is just a 0.2% advantage, while at 99 % the gap has widened to 7.4 %. On the other hand, for sparsity 80 % and 90 % the performance of Final-TopK is competitive with the other methods. This suggests that achieving these sparsity levels is not challenging on FEMNIST.

### 4.2.1 Zero-Sum Control Variates

In Section 3.4 we have demonstrated that if $\sum_i h_i \neq 0$, the algorithm diverges. This crucial observation is at the basis of our proposed method. We empirically confirmed on logistic regression for FEMNIST that this condition is violated for Accelerated-Server-Pruning and that this property leads to impaired performance on real world datasets. Details are found in Appendix D. To summarize, first, one can see that $\|\sum_i h_i\| > \frac{1}{N} \sum_i \|h_i\|$ showing that the sum is substantially far away from 0. Second, one can see that the algorithm converges to a substantially different solution as $\|w\|$ differs substantially between Accelerated-Server-Pruning and all other algorithms. The same holds for the norm of the gradient. To furthermore test the affect for real world test accuracy of FEMNIST, we can come up with a modified variant of Accelerated-Server-Pruning which keeps this conditions

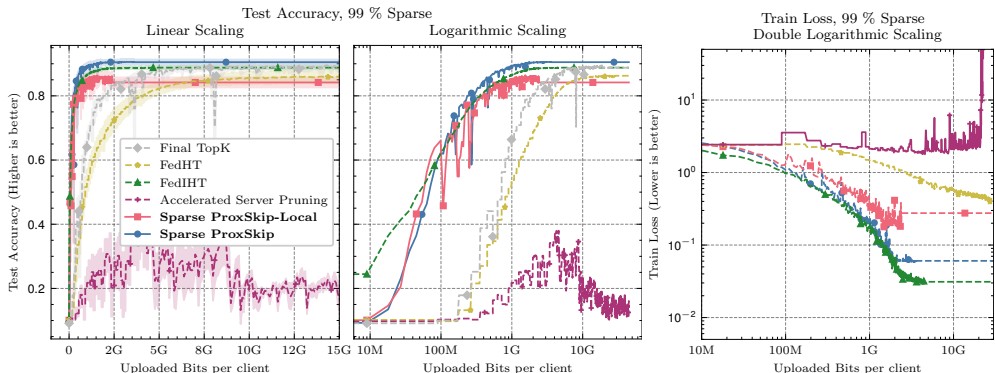

Figure 4: Results for ResNet18 (He et al., 2016) on CIFAR10 (Krizhevsky, 2009) at 90% sparsity. Sparse-ProxSkip is still able to outperform the baselines, although to a lesser degree. The main observation is that Accelerated-Server-Pruning fails completely in accuracy and loss because of $|\sum h_i| \gg 0$ and that the proposed fixes of Sparse-ProxSkip address this problem. The shaded area in the figures represents the standard error.

(as Accelerated-Server-Pruning does not) and Sparse-ProxSkip which violates this condition (as it always holds for Sparse-ProxSkip). We can see in Figure 6 that in both cases, the variant that keeps $\sum_i h_i = 0$ outperforms its counterpart substantially.

## 4.3 DEEP LEARNING EXPERIMENTS

Further nonconvex experiments were conducted on CIFAR10 (Krizhevsky, 2009) using ResNet18 (He et al., 2016). Further details can be found in Appendix E.

One can observe the results for 90% sparsity in Figure 4. Mainly, we note that Accelerated-Server-Pruning fails completely both in accuracy and in the loss increasing instead of decreasing. The algorithm does not head towards a minimum of the loss. This is because early on, the sum of the control variates $\sum h_i$ grows quickly and shifts all subsequent local gradients. Hence, one can see that keeping $\sum h_i = 0$ is particularly important for large models. Furthermore, one can see that the proposed variant Sparse-ProxSkip performs best and gives the highest final accuracy. We attribute the higher final accuracy to the control variates counteracting client drift. On the other hand, in this scenario we do not see a benefit from acceleration. This aligns with earlier observations that acceleration faces challenges in deep learning (Defazio & Bottou, 2019) and that addressing client drift proves beneficial for final accuracy nonetheless (Li et al., 2023). However, Li et al. (2023) found that control variates also benefit to the communication cost in highly heterogenous settings. We thus hypothesize that our setting was not heterogenous enough. While we applied the same federation process as Li et al. (2023), the different observation might be due to full client participation and a small number of clients in our experiments. In this setting, the amount of data per client is large and the heterogeneity of the Dirichlet distribution with parameter $\alpha$ might not lead to the same level of heterogeneity.

## 5 CONCLUSION

We investigated whether it is possible in FL to combine the recent theoretical advances of acceleration and client drift mitigation via local training, with sparse training. We showed that 1) the naive combination of these techniques fails; 2) it is theoretically and empirically crucial to keep the sum of the control variates to zero; 3) pruning should be done at the clients, not the server. Based on these important findings, we developed a theoretically-motivated method, Sparse-ProxSkip, which integrates the successful mechanism of TopK for sparse training in FL. Our experiments confirm its efficiency.

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

CONTENTS

Table 5: Blog Feedback Dataset results. Results were tuned for $\gamma$ and $p$ and hence show the improved scores due to addressing client drift.

| Sparsity | | 80 % | | 90 % | | 95 % | |
|---|---|---|---|---|---|---|---|
| | | Train Loss | Test $R^2$ | Train Loss | Test $R^2$ | Train Loss | Test $R^2$ |
| Existing | Final-TopK | $2.817e7$ | 26.4% | $2.877e7$ | 23.8% | $3.113e7$ | 16.4% |
| | FedHT | $3.056e7$ | 18.0 % | $2.951e7$ | 21.9 % | $3.288e7$ | 12.8 % |
| | FedIHT | $3.143e7$ | 16.5 % | $2.937e7$ | 22.4 % | $3.267e7$ | 12.3 % |
| | Accelerated-Server-Pruning | $2.872e7$ | 25.9 % | $2.991e7$ | 20.4 % | $3.217e7$ | 16.3 % |
| | RandProx-$l_1$ | $2.823e7$ | 26.3 % | $2.894e7$ | 24.1 % | $3.073e7$ | 18.8 % |
| Ours | Sparse-ProxSkip-Local | $2.818e7$ | 27.0 % | $2.856e7$ | 26.8 % | $2.938e7$ | 23.9 % |
| | Sparse-ProxSkip | $2.810e7$ | 26.7 % | $2.831e7$ | 27.1 % | $2.897e7$ | 26.7 % |

## A   GENERAL EXPERIMENTAL DETAILS

Our experiments were implemented in Python using Pytorch. We will release the code publicly upon acceptance of this paper. The experiments were conducted on our local workstations equipped with Intel(R) Xeon(R) Gold 6226R CPUs (2.90 GHz), 1 TB of RAM, and four Nvidia A100 GPUs, each with 40 GB of VRAM, although much less is required to reproduce these results. Each single training run of the experiments took no more than 20 hours of compute time. Some methods do not produce models at the desired sparsity, e.g. FedIHT usually yields a model of $70 - 90\%$ when given a target sparsity of $90\%$. Hence, before any evaluation of any method the models are pruned to the target sparsity by applying $\text{Top}K$.

## B   EXPERIMENTAL DETAILS: LINEAR REGRESSION

**Blog Feedback Dataset Details.** The dataset contains a number of blog posts with their respective number of comments so far and the goal is to predict the number of comments over the following 24h time window. For federating the dataset, it has a natural split by considering the source page where a particular blog post appeared, i.e. the website domain where it was published. For each domain, we create one client.[1] Furthermore, before federating we scale all attributes to be in the range $[0, 1]$ to make the computations more amenable. This results in a dataset with 554 clients. A histogram of the client size can be found in Figure 5 in the appendix. To add a bias term, which is usual for regression, we modify every sample to have an additional entry 1.

**Objective Function.** We optimize the objective function

$$f(w) = \frac{1}{N} \sum_{i=1}^{N} f_i(w) = \frac{1}{N} \sum_{i=1}^{N} \left( \frac{1}{2} \|A_i w - b_i\|_2^2 + \frac{\alpha}{4} \|w\|_2^2 \right) + \frac{1}{2N} \phi(w).$$

Here $\phi$ encodes our sparsity constraint, i.e. either $\|\cdot\|_1$ or cardinality constraints resulting in $\text{Top}K(\cdot)$ and $A_i$ is the local data matrix. $\alpha = 10^3$ in our experiments and was empirically chosen to give good $R^2$ on a validation set.

**Evaluation Metrics.** In addition to reporting the loss, the BlogFeedback dataset Buza (2013) contains a train and a test split. The test split is *out-of-distribution* which in this case means that the test data was recorded at least 1 month up to a year later compared to the training dataset. To measure the error for regression it is usual to report the $R^2$ metric which lies between 0 and 1 for favorable predictors. A $R^2$ value of 0 does not explain the dataset at all while a values of 1 would explain the dataset fully. Hence, a higher $R^2$ is better.

**Initialization.** Regression is a convex scenario, so that for RandProx convergence is guaranteed from any starting point. Thus, to induce sparsity from the beginning, the initial model is chosen as $w_{i,0} = \mathbf{0}$ for every $i$.

---

[1]In practice this means grouping by the first 50 columns as these are attributes of the source website and creating a client for each unique combination of values in these columns

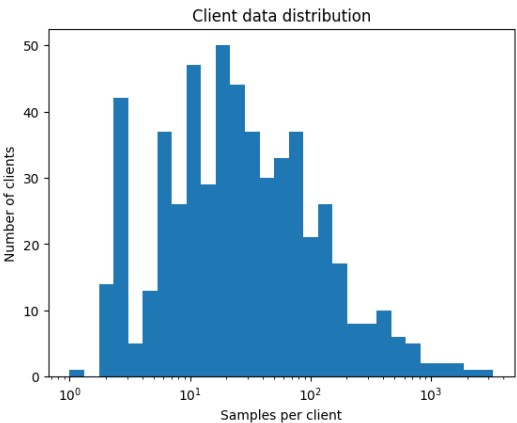

Figure 5: Distribution of the client sizes in the Federated version of the Blog Feedback dataset (Buza, 2013).

**Hyperparameters.** The hyperparameters, which are the learning rate $\gamma$ and average number of local steps $\frac{1}{p}$ were tuned by a random search. First a suitable range for these parameters was identified, then in a second random search the best parameters in this range were taken for the final experiments. Then, the average of 5 runs was taken to obtain the presented results. All algorithms were run for $10^4$ communication rounds ensuring convergence to their respective solutions.

**Full Experimental Results.** The results for the sparsity comparison including the loss function can be found in Table 5. From the loss one can see that the optimizer is not only better at increasing $R^2$, but also at decreasing the objective function.

## C EXPERIMENTAL DETAILS: LOGISTIC REGRESSION

**Dataset.** We run the experiments on the FEMNIST dataset (Caldas et al., 2018a), a common benchmark of the FL community that possesses a natural federated partition. We only consider $N = 100$ clients out of the 3220 naturally occurring in FEMNIST for the following reasons. A similar approach was taken by Jiang et al. (2022) for $N = 193$. On the one hand, ProxSkip requires modifications to support partial client participation (Condat et al., 2023; Grudzień et al., 2023), but in the setup chosen here only allows for full client participation. A high number of clients participating in each round is unrealistic (Charles et al., 2021). The goal of this work is to benchmark the advantage of control variates for client drift, hence providing a benchmark on natural federated splits is crucial. Merging clients would diminish the advantage of having a realistic federated split.

On the other hand, too few clients result in too little data. Hence, 100 was chosen as a tradeoff between these aspects resulting in a dataset of $n = 11152$ images. We employed the standard unrestricted test dataset. The performance tradeoff for this choice is that our centralized dense estimator achieves an accuracy of $89.4\%$ when trained on the full FEMNIST dataset, compared to $85.4\%$ when trained on our restricted dataset.

**Objective Function.** We align our objective function with the one from `scikit-learn` which uses the softmax formulation; that is, we define

$$\hat{p}_k(\mathbf{x}_i) = \frac{\exp(\mathbf{x}_i w_k + w_{0,k})}{\sum_{l=0}^{K-1} \exp(\mathbf{x}_i w_l + w_{0,l})}$$

and minimize

$$\min_w f(w) = \frac{1}{N} \sum_{i=1}^{N} f_i(w) = \frac{1}{N} \sum_{i=1}^{N} \left( -\frac{N}{n} \sum_{i=1}^{n_i} \sum_{k=0}^{K-1} [y_i = k] \log(\hat{p}_k(\mathbf{x}_i)) + \frac{\alpha}{2} \|w\|_2^2 \right) + \frac{1}{2N} \phi(\mathbf{w}).$$

$N$ is the number of clients, $n$ is the total number of samples and $n_i$ is the number of samples of Client $i$. Furthermore, $\mathbf{x_i}$ refers to a single datapoint and $y_i$ is its label.

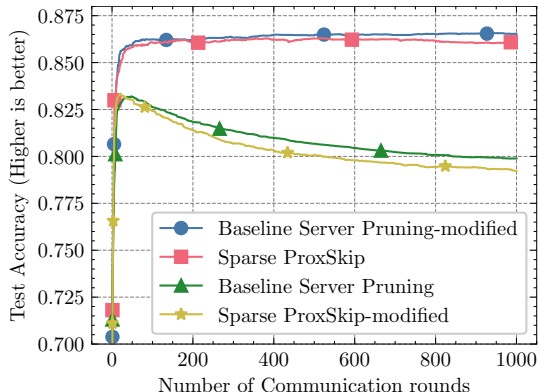

Figure 6: Test accuracy of our method and server pruning. The modified variants keep $\sum_i h_i = 0$. We can clearly see that this improves accuracy.

**Hyperparameters.** The hyperparameters of the learning rate $\gamma$ and local steps $\frac{1}{p}$ were tuned by a random search. First a suitable range for these parameters was identified, then in a second random search the best parameters in this range were taken for the final experiments. Then, the average of 5 runs was taken to obtain these results. The default initialization for a linear layer of Pytorch was taken.

## D    ZERO-SUM OF THE CONTROL VARIATES

This section provides empirical insights on why the property $\|\sum_i h_i\| = 0$ is crucial and its violation in Accelerated-Server-Pruning on logistic regression with FEMNIST and 90% sparsity. This refers to the setting and reasoning of Section 4.2.1.

First, Figure 6 shows the observation that Sparse-ProxSkip outperforms Accelerated-Server-Pruning. As a first step we introduce the following modified variants of these two algorithms. Sparse-ProxSkip-modified changes Line 9 of Algorithm 1 to be

$$w_{i,t+1} = \text{Top}K(\hat{w}_{i,t+1})$$

instead of

$$\hat{w}_{i,t+1} = \text{Top}K(\hat{w}_{i,t+1}).$$

This has the effect of potentially violating $\sum_i h_i = 0$. Furthermore, Accelerated-Server-Pruning-modified just switches Line 11 with Line 12 of Algorithm 1. This has the effect of fixing Accelerated-Server-Pruning to guarantee $\sum_i h_i = 0$. Figure 6 shows that the latter is a competitive variant and fixes the issue with Accelerated-Server-Pruning. Practically though, it is not very useful. It would require the full model to be sent to the models before they prune it locally. This saves neither on uplink nor downlink communication through compression.

First, on the left in Figure 7 one can see that $\sum_i h_i$ is far from 0, and combined with the plot on the right on the average norm of $h_i$, one can draw the conclusion that the size of $\sum_i h_i$ dominates the control variables themselves. Hence, with the proof from Section 3.4 one can conclude that the algorithm diverges by shifting the gradient by $\sum_i h_i$. To see this empirically, one can look at the norm of the parameters in Figure 8. Both Sparse-ProxSkip and Accelerated-Server-Pruning-modified converge to roughly the same parameters norm. The other variants though, for which $\sum_i h_i \neq 0$ holds, seem to move far away from this parameter combination. The plot on the left in Figure 8 confirms this in the loss: instead of minimizing the loss, the methods diverge significantly.

## E    EXPERIMENTAL DETAILS: DEEP LEARNING ON CIFAR10

**Experimental Details.** The experiments were run on CIFAR10 (Krizhevsky, 2009) using ResNet18 (He et al., 2016). The number of clients was $N = 10$ with full client participation. The data was distributed through a Dirichlet distribution with parameter $\alpha = 0.3$. The number of

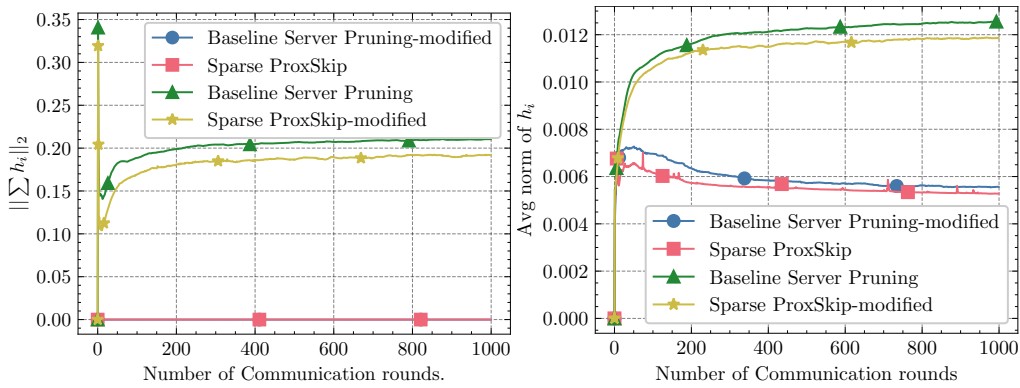

Figure 7: Norm of $\sum_i h_i$ on the left vs average norm of $h_i$ on the right.

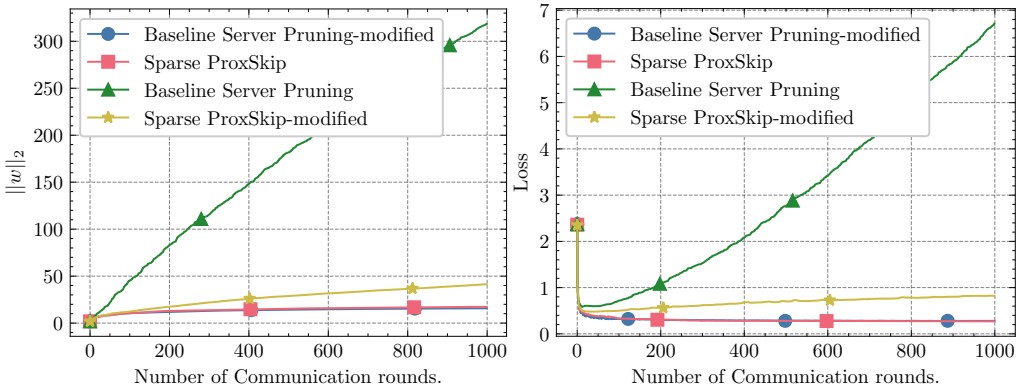

Figure 8: Norm of the model $w$ and loss value.

samples per client is distributed according to a lognormal distribution with variance 0.3. We used *FedLab* for producing the federated data split (Dun Zeng & Xu, 2021). A random search was conducted to find the best parameters among learning rate, local steps, batch size and gradient clipping value. The experiments were run for 500 rounds for. The number of local steps was chosen from the range $\{8, 16, 32, 64, 128, 256\}$. For ProxSkip, $p = \frac{1}{\#\text{local steps}}$ is taken. The batch size was chosen from the range $\{32, 64\}$. The gradients were clipped by a value chosen log-uniformly between 10 and 200. Without gradient clipping, ProxSkip would run into NaN errors. We used a weight decay of $10^{-4}$ and applied common transforms on the training data of flipping, cropping and normalizing.

## F  OUTLOOK AND LIMITATIONS

Sparse training might prove crucial for training large models in FL, which offer architectural benefits over small models. Here, sparse training enables larger models to respect the resource requirements of edge devices. Furthermore, these findings might be invaluable for combining centralized sparse training and pruning methods with acceleration. We provided a general invariant that pruning has to take place at the clients but future work might address the details of this integration. Additionally, in its current form, the method provides inference benefits and communication cost savings but would need further development for reducing the computational costs during training. In particular, our current gradients and control variables are dense, requiring further modification before yielding a sparse-to-sparse training method with the computational and memory footprint of a small model. In the pruning literature, masking is usually employed for this aspect. Here, one could apply masking to the control variates as well and combine gradient calculation and pruning as to decrease the memory cost of the full gradients.

