# OpenReview forum: "Prune at the Clients, Not the Server: Accelerated Sparse Training in Federated Learning"
_ICLR.cc/2025/Conference — ICLR 2025 Conference Withdrawn Submission_

### Official Review · Reviewer_Yjgp · 2024-10-31

**Soundness:** 2
**Presentation:** 3
**Contribution:** 2
**Rating:** 5
**Confidence:** 3

**Summary:**

This paper investigated the problem of sparse-to-sparse training in the non-convex setting and proposed an algorithm called Sparse-ProxSkip and provide experiments results on logistic regression on FEMNIST and  CNN on CIFAR-10.

**Strengths:**

The paper used $\ell_0$ penalty in RandProx and compared with different baselines in detail. The paper tried to show its performance boost on the CNN, which is non-convex problem.

**Weaknesses:**

This paper is entirely empirical, and I'm unclear on why the author addressed the proposed algorithm **can address** the non-convex problem when other algorithms cannot.
- Which specific modifications aim to tackle the non-convexity? Does this refer to the $l_0$ penalty? But the $l_0$ penalty is also applicable in convex optimization, such as in Lasso. This is supported by Figure 2, which shows that the proposed algorithm outperforms RandProx-$\ell_1$.
-  The original RandProx algorithm only ''achieves linear convergence results in the presence of strong convexity,'' which the $l_0$ penalty does not satisfy.
\end{itemize}

**Experiments**
- The tested sparsity levels are quite high, but the performance differences among algorithms seem to shrink at lower sparsity levels. What about the performance at sparsity levels of 20, 40, and 60?
- Given that the Baseline Server Pruning-modified algorithm appears to perform well, why not include it as a baseline comparison?

**Computational Cost:** Communication is a significant bottleneck. Considering that local devices are often mobile, the computational cost should also be evaluated. How much slower is the proposed algorithm on local devices compared to other devices?

**Questions:**

See weaknesses.

---

### Official Review · Reviewer_VdK5 · 2024-11-04

**Soundness:** 3
**Presentation:** 4
**Contribution:** 3
**Rating:** 5
**Confidence:** 4

**Summary:**

This paper explores a new approach to improve computational efficiency in Federated Learning (FL) through sparse model training. The authors propose an algorithm called Sparse-ProxSkip, which enables sparse-to-sparse training directly on clients rather than at the central server. Existing methods, such as pruning at the server, fail to combine acceleration and sparsity effectively. Sparse-ProxSkip offers a solution by focusing on sparse training at the clients, improving both computational efficiency and communication costs.

**Strengths:**

1. Sparse-ProxSkip addresses key obstacles in FL by integrating sparse-to-sparse training with local pruning, which is particularly well-suited for resource-constrained environments.

2. Sparse-ProxSkip outperforms baselines across various tasks, particularly in settings where high communication efficiency is essential, demonstrating its practical applicability.

**Weaknesses:**

1. In Algorithm 1 and the experiment setup, only full client participation is considered, which could limit the applications of the proposed method in practice.

2. In Algorithm 1, only the communication cost is reduced. The computational cost and memory cost is not saved during local update, (the parameter zeroed by TopK can be non-zero again) which is not well-suited for devices with limited computational resources.

3. The pruning technique in this method is TopK. In the deep learning experiment, it is not compared with dropout or other pruning methods that reduces the number of nodes in deep neural networks (e.g. comparing the accuracy, computational cost).

**Questions:**

Please see the weakness.

---

### Official Review · Reviewer_EJnV · 2024-11-04

**Soundness:** 3
**Presentation:** 3
**Contribution:** 2
**Rating:** 5
**Confidence:** 3

**Summary:**

This paper proposes a novel FL method that particularly combines sparse-to-sparse (STS) training with accelerated local optimization and is foundationally based on the idea of Prox-Skip and Rand-Prox to enhance sparse training in FL with acceleration. It relies on control variates to manage client drift. The paper is nicely formulated and well written, and I liked the presentation. Experimental results demonstrate the effectiveness of the method.

**Strengths:**

1)	Smart adaptation of sparsity with acceleration.
2)	The experiments on higher sparsity levels and reported communication speedup is something to be appreciated.
3)	I like the way paper is written, should be easily understandable to the readers.

**Weaknesses:**

1)	I agree that convergence is theoretically guaranteed in convex scenarios, but it is not guaranteed for nonconvex models resulting in lower stability, particularly for highly nonconvex scenarios. I recommend revising the abstract to avoid overstating its applicability to nonconvex scenarios. A more accurate phrasing could be: "Developed for convex scenarios, though potentially useful for nonconvex settings," reflecting the current scope of theoretical guarantees.
2)	Although it is a smart adaptation, I would still consider the novelty to the lower end.
3)	Although communication speedup, I am still concerned about the computational costs as it still requires the dense gradients and control variates.
4)	Baselines although few and validated through experiments on core works on the foundation of the proposed method, I believe considering a few more recent baselines is a must for a thorough analysis. Recent pruning at client sparse baselines include DisPFL[1], SSFL[2], etc.
5)	Another major concern is the model size experiments. Only ResNet18 is used, which is a smaller model. Sparse models in FL could behave differently with model size, considering increasing complexity. For an effective contribution, consider experimenting on at least one larger model (ResNet 32 or above).

References:

[1] https://doi.org/10.48550/arXiv.2206.00187

[2] https://doi.org/10.48550/arXiv.2405.09037

**Questions:**

1. Please see questions on weaknesses.
2. My opinion is that this idea should work (according to experiments), however the paper is lacking rigorous experiments and is purely focused on experimenting on the foundational work this idea is based on. Can authors convince me otherwise? Overall, I believe a paper has potential, but this version seems a little too narrowly experimented and grounded.

---

### Official Review · Reviewer_EZPw · 2024-11-04

**Soundness:** 3
**Presentation:** 3
**Contribution:** 2
**Rating:** 3
**Confidence:** 4

**Summary:**

This paper addresses the high communication costs inherent in federated learning and proposes a sparse training approach on the client side for nonconvex settings. The approach combines the ProxSkip and RandProx methods and introduces a novel method for determining control variables to achieve model sparsity, thereby reducing communication overhead effectively.

**Strengths:**

1. The paper proposes various methods to address the placement of the top-k operator, analyzing and comparing the performance of each approach.
2. The authors introduce and validate a crucial criterion to ensure model performance, specifically that $\sum_i h_i=0$.
3. Extensive experiments are conducted to demonstrate the effectiveness of the proposed methods.

**Weaknesses:**

1. The paper primarily integrates existing methods to create a new approach rather than developing a novel method from the ground up.
2. The computational complexity of the top-kk operation is not addressed. Top-k selection can demand significant computational resources and time, which may render it impractical for client devices, particularly when dealing with large model parameters. This issue is especially relevant given the limited computational power of clients compared to a central server.
3. The distinction between Sparse-ProxSkip-Local and Sparse-ProxSkip is unclear, as both appear to follow the same steps in Algorithm 1.  Additional clarification is necessary to outline any differences between these methods.
4. More explanation is required regarding the control variables $h_i$. Specifically, in Sections 3.2 and 3.3, $h_i$  appears as a loss function, while in Section 3.4, it is treated as a variable. It is crucial to clarify the role of $h_i$ in the algorithm, along with the significance of the criterion $\sum_i h_i=0$. It is important to figure out why using $h_i$ in the algorithm and what does $\sum_i h_i=0$ mean.
5. What challenges does the nonconvex setting present compared to existing convex methods, and how are these challenges addressed?

**Questions:**

Please refer to the Weakness.

---

### Official Review · Reviewer_wAEk · 2024-11-06

**Soundness:** 2
**Presentation:** 2
**Contribution:** 2
**Rating:** 3
**Confidence:** 4

**Summary:**

**Summary.** The authors propose Sparse-ProxSkip algorithm, an extension of RandProx (Condat & Richtarik, 2022), for training non-convex objectives that combines sparse training and acceleration for federated training. The authors conduct extensive experiments to demonstrate the performance of the proposed algorithm.

**Strengths:**

**Strengths.** Here, I list the strengths of the paper.
- The proposed algorithm presents an interesting application of RandProx and ProxSkip for training FL systems with sparse solutions.
- The authors showcase impressive numerical performance for the proposed algorithm.

**Weaknesses:**

**Weaknesses.** Here, I list some major issues with the paper.

- The major issue with the paper is that the developed algorithm looks to be a simple extension of RandProx (and ProxSkip) with the proximal operator substituted for a sparse operator. All in all, the contribution of the paper looks to be very limited since
    - The algorithm does not seem to be novel.
    - There are no theoretical guarantees or analyses for the non-convex setting considered in the paper.

- I do not understand the usage of the term "acceleration" throughout the paper. Usually, acceleration in optimization literature is referred to as Nesterov's acceleration wherein the iteration complexity of the algorithms is improved. However, I believe here there is no acceleration happening and the authors refer to acceleration as communication reduction.

- In both Steps 7 and 9 why the Sparse-ProxSkip step is required? Isn't it enough to apply Sparse-ProxSkip in Step 7 only?
- The writing of the paper can improve.

Overall, I believe that although the proposed algorithm leads to good numerical performance, the contribution of the paper is limited both algorithmically and theoretically.

**Questions:**

Please see the weaknesses above.

---

### Note · Authors · 2024-11-18

I have read and agree with the venue's withdrawal policy on behalf of myself and my co-authors.